human-computer interaction/biocomplexity/complexity

segregation, polarization, urban systems, data science, human behaviour

**Author for correspondence:**
Alfredo J. Morales
e-mail: alfredo@necsi.edu

# Segregation and polarization in urban areas

Alfredo J. Morales[1,2], Xiaowen Dong[2,3], Yaneer Bar-Yam[1] and Alex 'Sandy' Pentland[2]

[1]New England Complex Systems Institute, Cambridge, MA, USA
[2]MIT Media Lab, Cambridge, MA, USA
[3]University of Oxford, Oxford, UK

 AJM, 0000-0002-8509-0839; XD, 0000-0002-1143-9786;
YB-Y, 0000-0001-8830-9237; AP, 0000-0002-8053-9983

Social behaviours emerge from the exchange of information among individuals—constrained by and reciprocally influencing the structure of information flows. The Internet radically transformed communication by democratizing broadcast capabilities and enabling easy and borderless formation of new acquaintances. However, actual information flows are heterogeneous and confined to self-organized echo-chambers. Of central importance to the future of society is understanding how existing physical segregation affects online social fragmentation. Here, we show that the virtual space is a reflection of the geographical space where physical interactions and proximity-based social learning are the main transmitters of ideas. We show that online interactions are segregated by income just as physical interactions are, and that physical separation reflects polarized behaviours beyond culture or politics. Our analysis is consistent with theoretical concepts suggesting polarization is associated with social exposure that reinforces within-group homogenization and between-group differentiation, and they together promote social fragmentation in mirrored physical and virtual spaces.

## 1. Introduction

Urban segregation [1] is a complex process that does not require a centralized agent to enforce [2]. It can emerge due to the behaviour of individuals like other self-organizing properties of complex systems [3]. Group polarization [4] can also emerge from autonomous and distributed individual actions [5]. The choice of whom to interact with and imitate influences the information people receive and the way individuals think and behave, evolving toward self-reinforcing group identities. It is crucial to understand how the segregation of social interactions and constraints of information flows are related to the differentiation, and polarization, of emergent social behaviours.

The social space is where people gather and interact, both physically and online. Members of a social space share stronger relationships with each other than with people from separated spaces. The strength of relationships has a key role in collective learning and the spread of information and behaviours [6]. While people are mostly influenced by stronger and closer ties, weaker ties are responsible for integrating social groups at higher scales. Segregation directly affects the way people create and maintain strong ties and therefore, the properties and collective behaviours of the emergent social space.

Urban segregation has been traditionally analysed by mapping the geographical distribution of homes in urban areas using surveys or Census data [7]. Previous studies show that the racial and income distribution of neighbourhoods have historically shaped the evolution of American cities [8] at multiple scales [9,10]. Either directly through explicit policies or indirectly due to market behaviours, the infrastructure and development of American neighbourhoods and suburbs have in some cases reinforced the segregation of social groups with direct implications on economic mobility [11], environmental health [12] and democracy [13]. The migration of white and richer populations towards suburbs worsened the inner-city poverty and impeded the integration of poorer communities in society [8]. Many negative effects have been attributed to segregation. For example, the displacement of companies towards suburban developments affects the way people commute and therefore constrains the way information, money and opportunities flow in cities—particularly affecting inner-city poverty [11]. Moreover, the lower price of land in poorer areas has attracted polluting facilities that affect the health and life span of poorer communities [12]. Finally, jurisdictions prevent the equal representation of distinct social groups in decision-making processes when favouring affluent and economically powerful areas [13].

Data from electronic media has recently enabled unprecedented analysis of societies across scales of observation, from individual to collective behaviours [14–18]. Previous studies of urban segregation using geolocated Twitter data have shown that different ethnic groups are less exposed to each other because of segregated residential and travel patterns [19,20]. We analyse Twitter and credit card shopping data (see Methods) in order to understand the relationship between geographical and virtual segregation and their relationship to topics of communication. We consider virtual segregation as the preference of people to interact with those of similar demographics (i.e. income). We show that physical and online interactions across cultures are dominated by income and linked to differentiated and polarized collective interests and conversational domains.

## 2. Results

To analyse patterns of segregation, we build urban networks of social interactions. Nodes represent neighbourhoods, and edges indicate whether people interact with individuals in other neighbourhoods. Edges are *social bridges* connecting neighbourhoods and transferring information. Figure 1 shows the structure of three types of interaction networks: (i) shopping, (ii) human mobility, and (iii) communication on Twitter, representing Istanbul (figure 1a) and New York City (figure 1b). Results for five more cities are presented in figure 2. See Methods for a detailed description of each network. Human mobility refers to how individuals visit different neighbourhoods. Communication on Twitter refers to how individuals talk to other individuals that live in different neighbourhoods via the mentions mechanism. We aggregate neighbourhoods into $q = 10$ income quantiles that represent the axes of each matrix (0 being the neighbourhoods with the lowest median income and 9 the highest). A table showing the income ranges per quantile is presented in electronic supplementary material, S1. The vertical axis indicates the target of the interaction and the horizontal axis represents the source. Each element of the matrix, therefore, represents the interaction between a certain pair of income quantiles. We first normalize each element of the raw interaction matrix by dividing it by the sum of the corresponding column, and then subtract the expected value for a uniform probability of interaction ($p_u = 1/q$). As a result, each column shows the tendency of the source of an interaction conditioned on its target. Values above $p_u$ before subtraction are coloured in red and below in blue. The red blocks along the diagonal of the matrices in figure 1 indicate a strong segregation of social interactions by income. People interact primarily with their own socio-economic group. Segregation is much more pervasive than generally thought; it does not just separate the wealthy from the poor, but more granularly between socio-economic classes. Moreover, the blue upper right and lower left corners show that the highest and lowest income quantiles have the least interaction.

In order to measure the amount of segregation on these networks, we compare the distributions of the source of interactions conditioned on their target (matrix columns in figures 1 and 2) with the

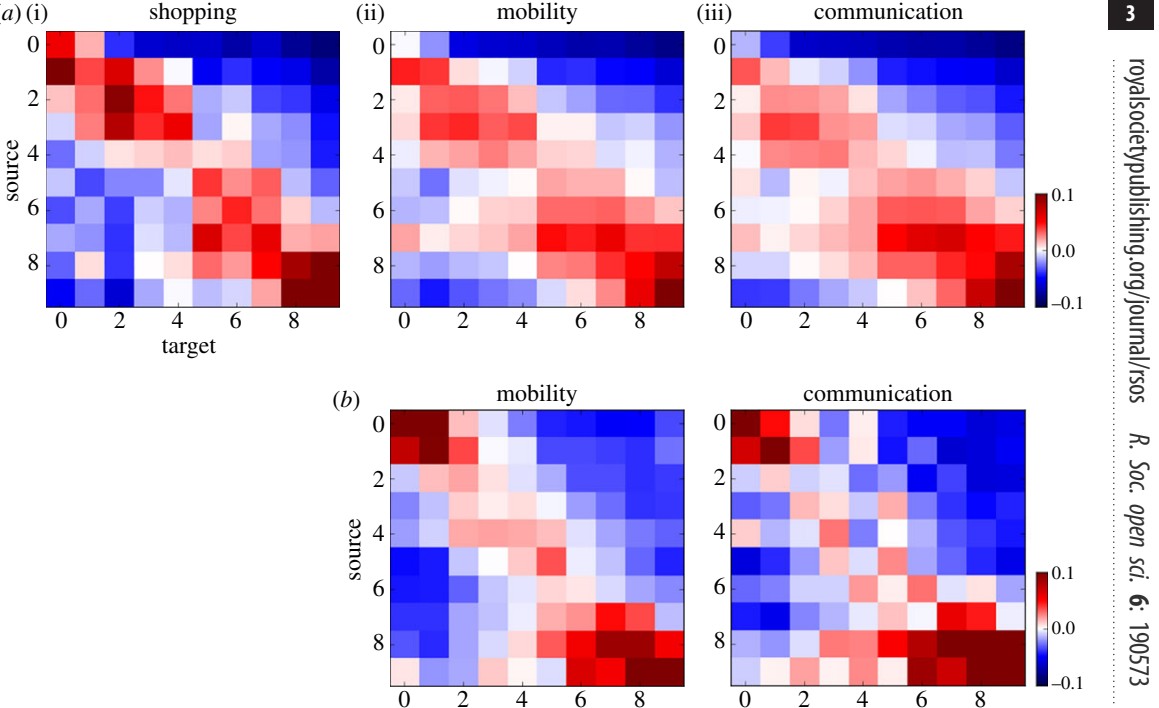

**Figure 1.** Interaction matrices by type of activity for Istanbul (*a*) and New York City (*b*). Each matrix shows the normalized sum of interactions among neighbourhoods according to shopping data (i), mobility based on Twitter (ii) and communication on Twitter (iii). Neighbourhoods are aggregated into ten income quantiles numbered from 0 (lowest median income) to 9 (highest median income) on the axes. Columns have been normalized by their sum and the probability of a random connection assuming a uniform distribution among quantiles has been subtracted. Blue regions show areas with connections below random values and red areas show connections above. Shopping data are only available for Istanbul. Results for more cities are presented in figure 2.

uniform distribution. In figure 3, we show the Q-Q plots of the empirical distributions against the uniform one. There is one plot per network and one curve per economic quantile (colour) in each plot. If interactions were not segregated, the curves should closely follow the diagonal line. Concave down curves indicate a preference of interactions with lower quantiles. Concave up curves indicate a preference of interactions with wealthier quantiles. The area wrapping the curves is proportional to the amount of segregation on the network. The value of the areas are presented in table 1. Mobility networks are consistently less segregated than communication networks. Cities like Istanbul and Los Angeles present less segregation than other cities in terms of mobility. A set of Kolmogorov–Smirnov statistical tests [21] applied to these curves show that the conditional distributions significantly differ from a uniform probability distribution across all networks and cities ($p < 0.001$, see electronic supplementary material, S2.1). Moreover, balanced neighbourhood samples proportional to the neighbourhood population according to Census data show consistent results, confirming the robustness of our observations (see electronic supplementary material, S2.2).

The systematic breakpoints in social communication affect the spread of information and, since we learn from imitation, may also promote divergence of behaviours [5]. The differentiation and polarization of behaviours due to the segregation of interactions is reflected in collective interests and topics of conversation on social media. We apply principal component analysis (PCA) [22] to human mobility and hashtag[1] usage matrices. The mobility PCA manifests the way people move around the city, while the hashtag usage PCA informs us about sharing of topics online. The PCA begins from matrices where rows represent neighbourhoods and columns represent either tweeting individuals or hashtags posted. The PCA decomposes the original data space into orthogonal vectors, named principal components, which characterize the structure of the system [23]. In the case of Istanbul and New York City, the component that explains most of the variance is well correlated with the neighbourhood income for both human mobility and hashtag usage ($r = 0.57$ and $r = 0.56$, respectively, in the case of Istanbul). A summary of the correlations of the principal components with income for

---

[1]Hashtags are keywords that people use to identify their tweets with ongoing trends and are a proxy of collective attention.

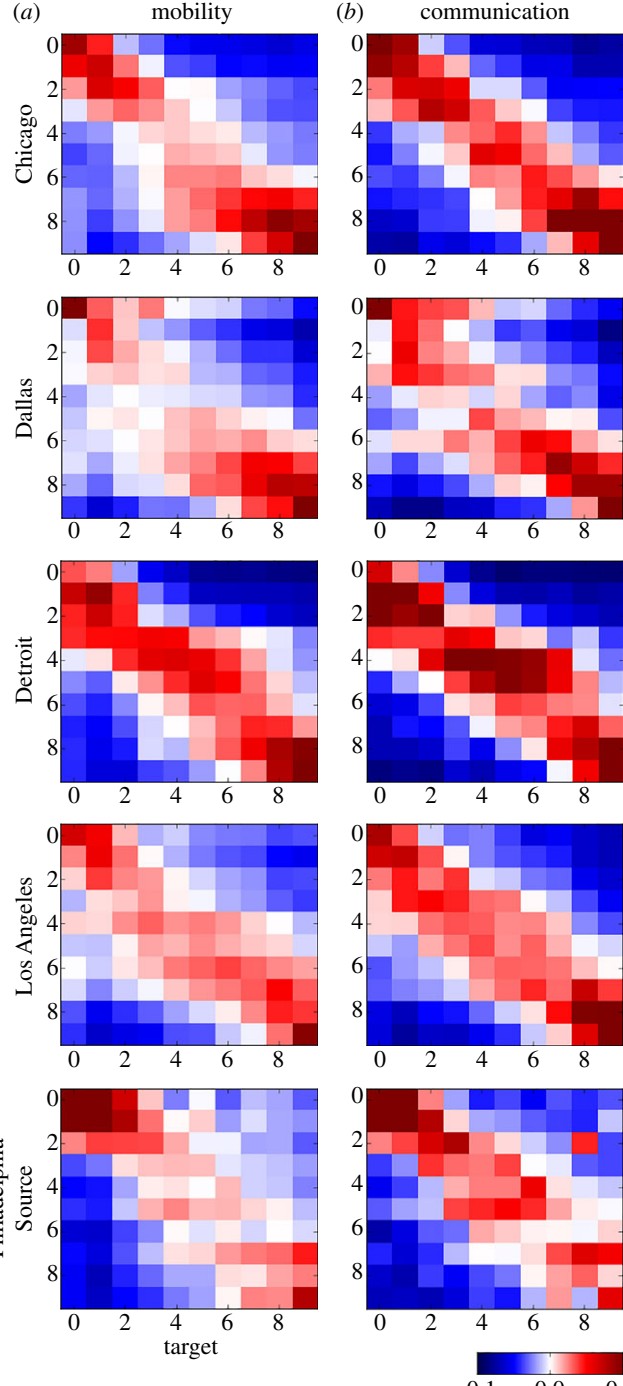

**Figure 2.** Interaction matrices by type of activity for Chicago, Dallas, Detroit, Los Angeles and Philadelphia, which show normalized sum of interactions between each pair of neighbourhoods according to mobility (a) and communication (b) on Twitter. Neighbourhoods are aggregated into $q = 10$ income quantiles represented on the axes. Blue regions represent the pairs of income quantiles where probability of interaction is below the expected value of a uniform distribution $p_u = 1/q$, and red ones above.

all cities is presented in table 2. In electronic supplementary material, S3, we present a full description of the spatial structure of the top 20 components for all cities. We also present an analysis of the statistical significance of the correlation of the main components with neighbourhood income and, in the case of Istanbul, also with political preferences.

Figure 4 shows a scatter plot of the coordinates of neighbourhoods' principal hashtag component (vertical axis) and mobility component (horizontal axis) for Istanbul and New York City. Results for five additional cities are presented in electronic supplementary material, S3. Dot colour represents

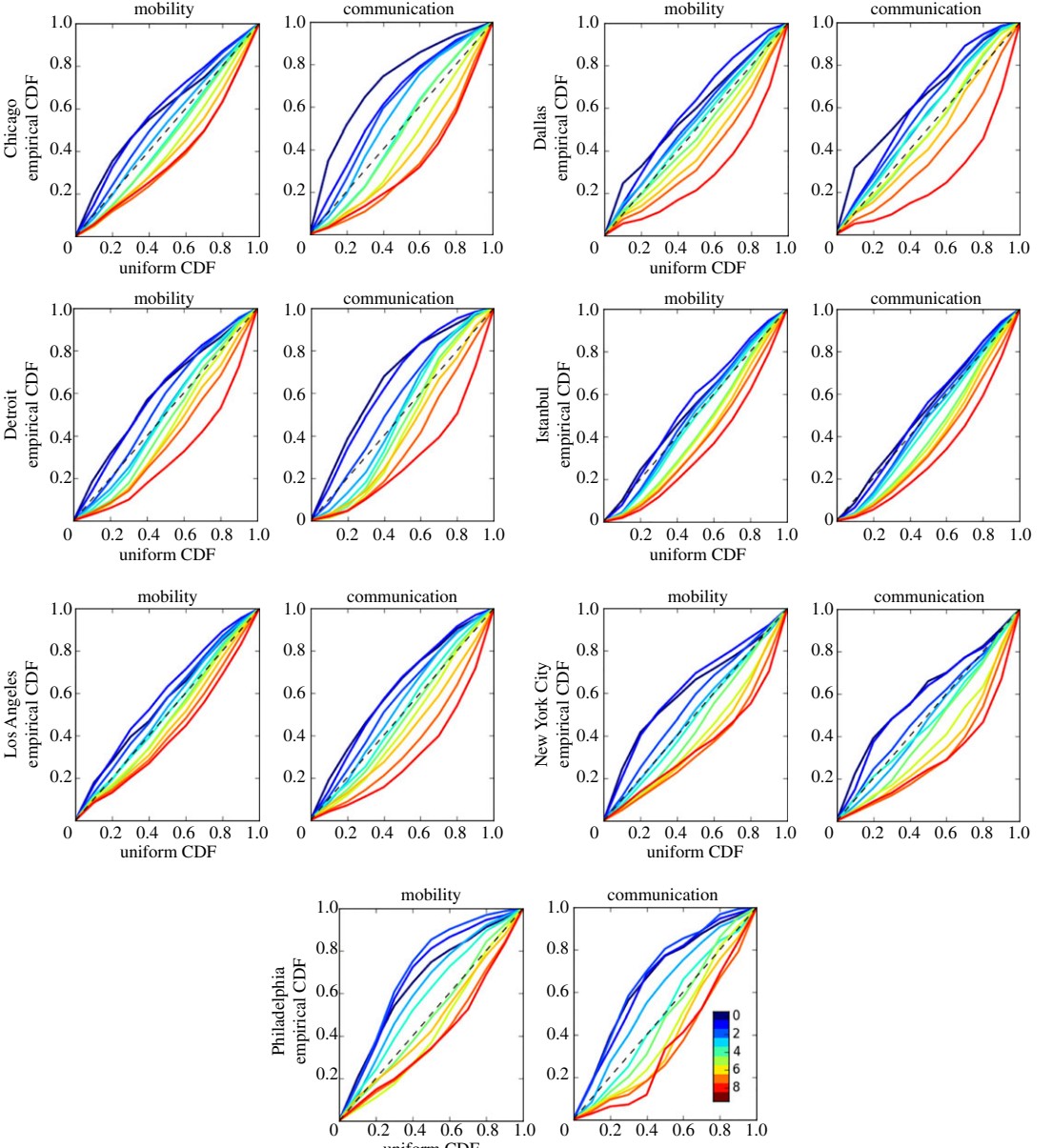

**Figure 3.** Q-Q (quantile-quantile) plots of the distributions of the source of interactions conditioned on their target (columns in matrices of figures 1 and 2) against the uniform distribution. There is one plot per network and one curve per quantile (colour) in each plot (scale inset). Concave down curves show a preference of interactions with lower quantiles. Concave up curves show a preference of interactions with wealthier quantiles. Curves near the diagonal show less segregation. The area wrapping the curves is proportional to the amount of segregation on the network. The value of the areas are presented in table 1. A set of Kolmogorov–Smirnov statistical tests [21] applied to these curves show that the conditional distributions significantly differ from a uniform probability distribution across all networks and cities ($p < 0.001$, see electronic supplementary material, S2.1).

neighbourhood median income. The apparent separation between richer (blue) and poorer (red) neighbourhoods shows that different income groups are distant both in the physical space and online. They are neither found in the same places, nor discussing similar issues. The differentiation of behaviours among isolated populations has been previously discussed in the sociological [24] and complex systems literature [5]. Our analysis uses large-scale human behaviour data and exposes income as a fundamental dimension of isolation in urban environments. Previous analyses based on surveys confirm that income drives the differentiation of social behaviours [25,26].

To analyse the polarization of conversations between neighbourhoods, we applied a topic model clustering algorithm [27,28] to the matrix of neighbourhoods' hashtags. The topic model reveals groups of hashtags that co-occur and can characterize topics of conversations. From a total of 20 hashtag clusters, we highlight two of them whose usage across neighbourhoods is positively and

**Table 1.** Segregation index based on the area between Q-Q plots comparing the empirical CDF with theoretical uniform one (figure 3). We present the segregation index for mobility and communication networks in all cities. See Methods for a detailed description of the networks.

| city | mobility | communication |
|------|----------|---------------|
| Chicago | 0.234 | 0.399 |
| Dallas | 0.314 | 0.384 |
| Detroit | 0.296 | 0.373 |
| Istanbul | 0.211 | 0.205 |
| Los Angeles | 0.185 | 0.313 |
| New York City | 0.288 | 0.308 |
| Philadelphia | 0.318 | 0.347 |

**Table 2.** Principal components used in figure 4 and electronic supplementary material, figure S2. The spatial structure of these components is shown in electronic supplementary material, figures S3–S16. The correlation of those components with income is presented in parenthesis ($r$) and shown in electronic supplementary material, figures S17 and S18. The correlations with income of the components shown in this table are significant ($p < 0.001$) as shown in electronic supplementary material, figure S19.

| city | hashtag | mobility |
|------|---------|----------|
| Chicago | PC-2 ($r = 0.43$) | PC-2 ($r = 0.41$) |
| Dallas | PC-4 ($r = 0.45$) | PC-2 ($r = 0.35$) |
| Detroit | PC-1 ($r = 0.39$) | PC-3 ($r = 0.43$) |
| Istanbul | PC-1 ($r = 0.56$) | PC-1 ($r = 0.57$) |
| Los Angeles | PC-3 ($r = 0.22$) | PC-2 ($r = 0.24$) |
| New York City | PC-1 ($r = 0.39$) | PC-1 ($r = 0.39$) |
| Philadelphia | PC-2 ($r = 0.47$) | PC-1 ($r = 0.50$) |

negatively correlated with the principal hashtag component ($r = 0.7$ and $r = -0.8$ for Istanbul) and neighbourhood median income ($r = 0.4$ and $r = -0.6$ for Istanbul), respectively. The spatial distribution of topic cluster usage is shown in figures 5 and 6. Two topics are presented for each city. One topic is popular in wealthier areas (Topic-1), while the other topic is popular in poorer areas (Topic-2). The scatter plots show that the popularity of topics is mutually exclusive among wealthier and poorer areas. In electronic supplementary material, S3, we present a set of statistical tests performed for all cities to measure the significance of the clusters found by comparing the empirical distributions with those obtained from a randomized hashtag space. All correlations and statistical properties of the randomized space are shown in table 3. The probability that the topics reported in these figures occur at random is extremely low across all cities.

The polarization of geographically central (figures 5a(i),b(i) and 6b) and peripheral (figures 5a(ii),b(ii) and 6a) neighbourhoods is manifested. We quantify the geographical extent of topics by calculating their average distance to the city centre. For this purpose, we (i) normalize the strength of each topic across neighbourhoods such that their sum equals 1 in each city, (ii) multiply the new weight by the distance between the neighbourhood centroid and the geographical city centre, and (iii) sum across neighbourhoods such that we obtain a weighted average topic distance per city. The results are presented in table 4. In Istanbul and New York City, the wealthier populations live closer to the city centre and thus the average distance of wealthier topics is lower than the average distance of poorer ones. Meanwhile, in Chicago, Detroit, Los Angeles and Philadelphia, the average distance of wealthier topics from the city centre is larger than the distance of poorer ones. Historical evidence explains the creation and worsening of inner-city poverty due to policies that either directly or indirectly promoted a geographical patterns of racial and income segregation [8]. We show that such distribution of wealth and people is reflected in the differentiation of online conversations.

Richer and poorer neighbourhoods do not seem to talk to each other and are interested in very different topics. For example, in Istanbul, richer neighbourhoods discuss lifestyle topics using English

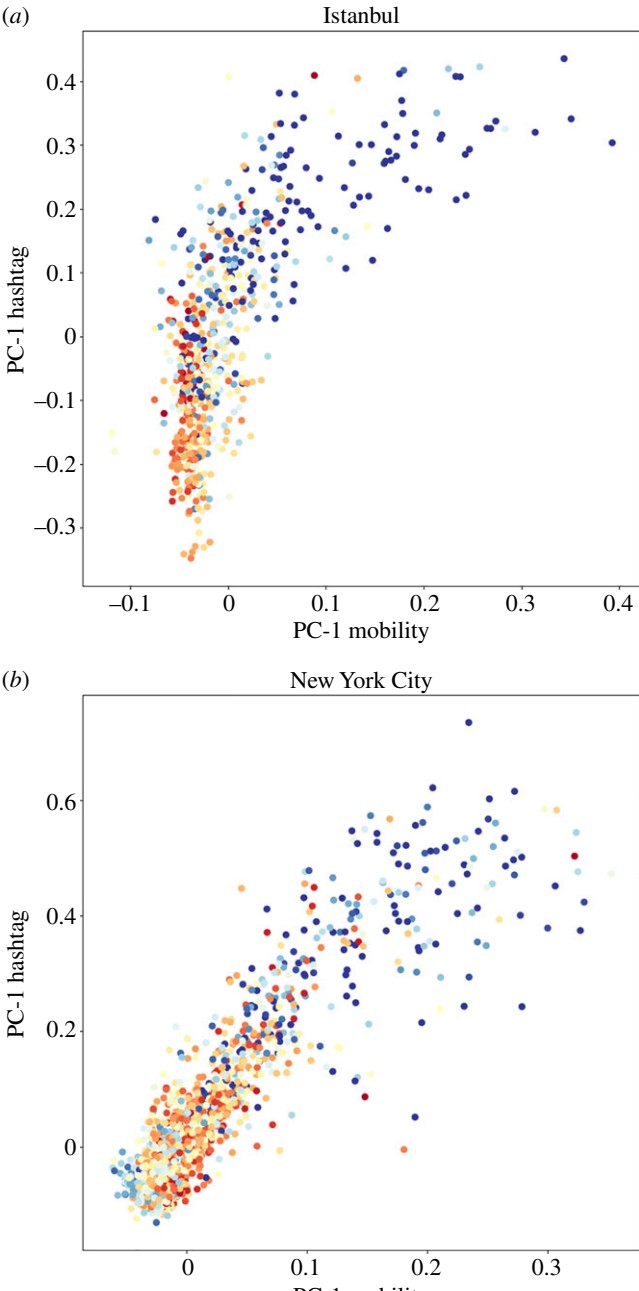

**Figure 4.** Principal component analysis (PCA) of hashtag usage (*y*-axis) and human mobility patterns (*x*-axis) for Istanbul (*a*) and New York City (*b*). Dots represent neighbourhoods and colours indicate the median income (from red to blue). Each axis represents the projection of the neighbourhood data onto the respective first principal component. Results for more cities are presented in electronic supplementary material, S3.1.

words like: party, friends and love; while poorer communities discuss religion, politics, TV shows and sports. In American cities, lifestyle hashtags also prevail in richer areas, while sports, zodiac signs and horoscopes seem to be more relevant in poorer areas. In electronic supplementary material, S3, we present a description of the hashtags used in richer and poorer areas and an analysis of the statistical significance of topic correlation with neighbourhood income for all cities. The high correlation between content polarization and segregation of interactions suggests that these two phenomena are linked to each other (figure 7).

We studied the hypothesis that urban segregation by income determines the structure of social bridges and therefore influences mutual exposure and its role in homogenization. Mutual exposure is measured by mobility matrices that count the number of people living in neighbourhood $i$ that visit neighbourhood $j$. Homogenization is measured through hashtag usage. We created matrices of hashtag similarities that

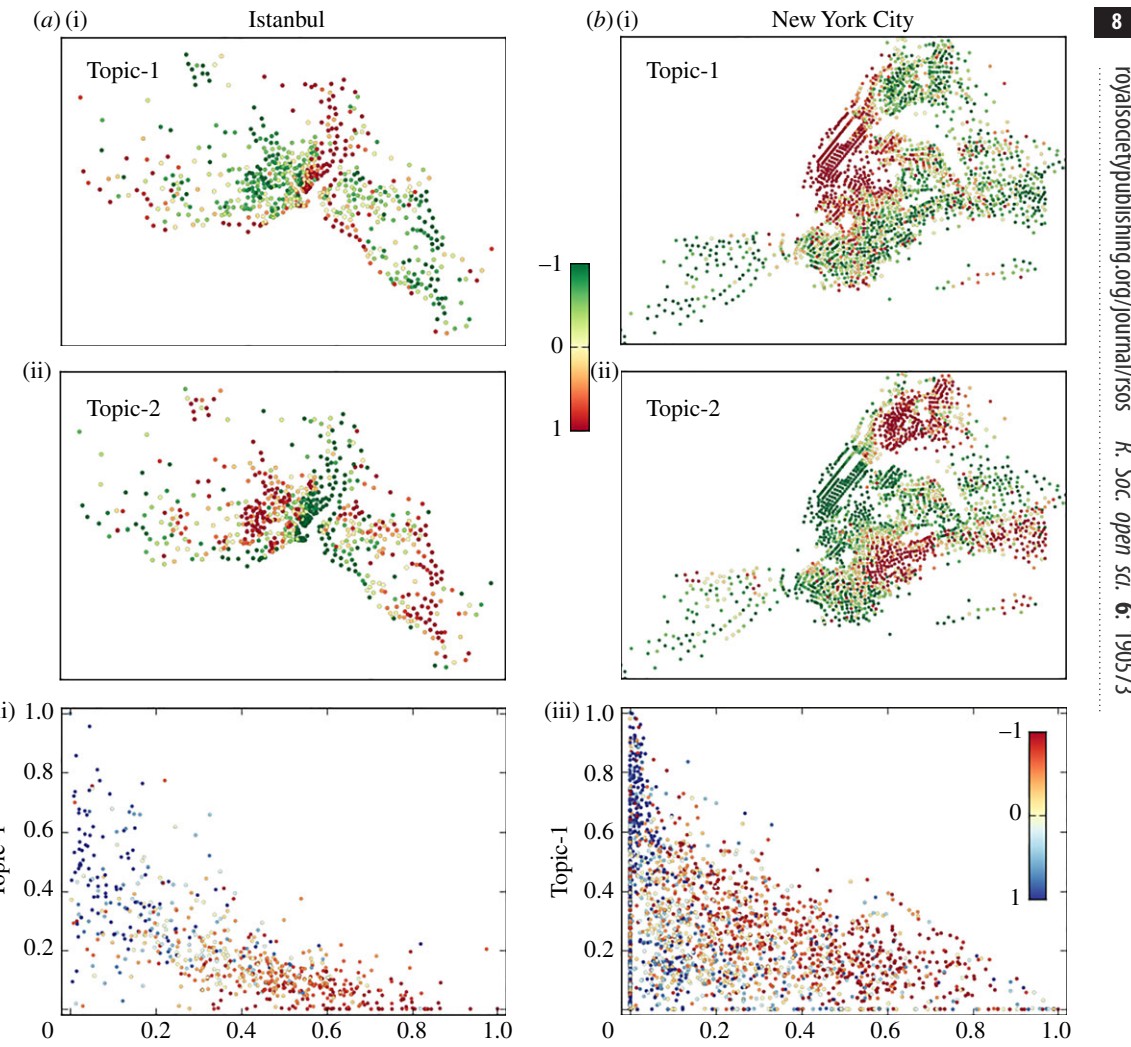

**Figure 5.** Polarization of discussion topics across neighbourhoods. Spatial distribution of topics on Twitter for Istanbul (*a*) and New York City (*b*). (*a*(i),*b*(i)) Dominant topics in wealthier areas (Topic-1). (*a*(ii),*b*(ii)) Dominant topics in poorer areas (Topic-2). Dots indicate geolocated neighbourhoods and colours indicate the normalized intensity of the cluster in each neighbourhood (colour scale shown in figure). (*a*(iii),*b*(iii)) Scatter plots in which each neighbourhood is represented by a dot and the coordinates represent the extent of conversation of Topic-1 (vertical axis) and Topic-2 (horizontal axis) in that neighbourhood. Dot colour indicates normalized median income (scale inset). Results for more cities are presented in figure 6.

measure the cosine distance between the hashtag vectors of each pair of neighbourhoods. The hashtag similarity matrices for Istanbul and New York City are presented in figure 7*b* (first and second row). They show a similar block structure across the diagonal like the communication and mobility matrices presented in figures 1 and 7*a*. The correspondence of the structure of these matrices indicates that the similarity of hashtag usage is consistent with the segregation of social interactions (figure 7*c*). Results for five additional cities are shown in electronic supplementary material, S6. Moreover, in the case of Istanbul, wealthier areas present more coherent hashtag vectors than poorer neighbourhoods (electronic supplementary material, S5) and human mobility is a better predictor of income quantile than hashtag usage (electronic supplementary material, S4).

We analyse the correspondence of structure in the mobility and hashtag similarity matrices by means of pair matching and aggregation [29]. We aggregate neighbourhoods both by income and random association. Figure 7 shows mutual exposure (via mobility) and behavioural homogenization (via hashtag similarity) between different income groups (first and second row), together with that between randomized groups (third and fourth row), in Istanbul and New York City. Additional cities are shown in electronic supplementary material, S6. Aggregating by income presents clear patterns of segregation in both measures, while the randomized matrices are homogeneous and uniform. Regardless of the type of aggregation, both matrices are very well correlated with each other (scatter

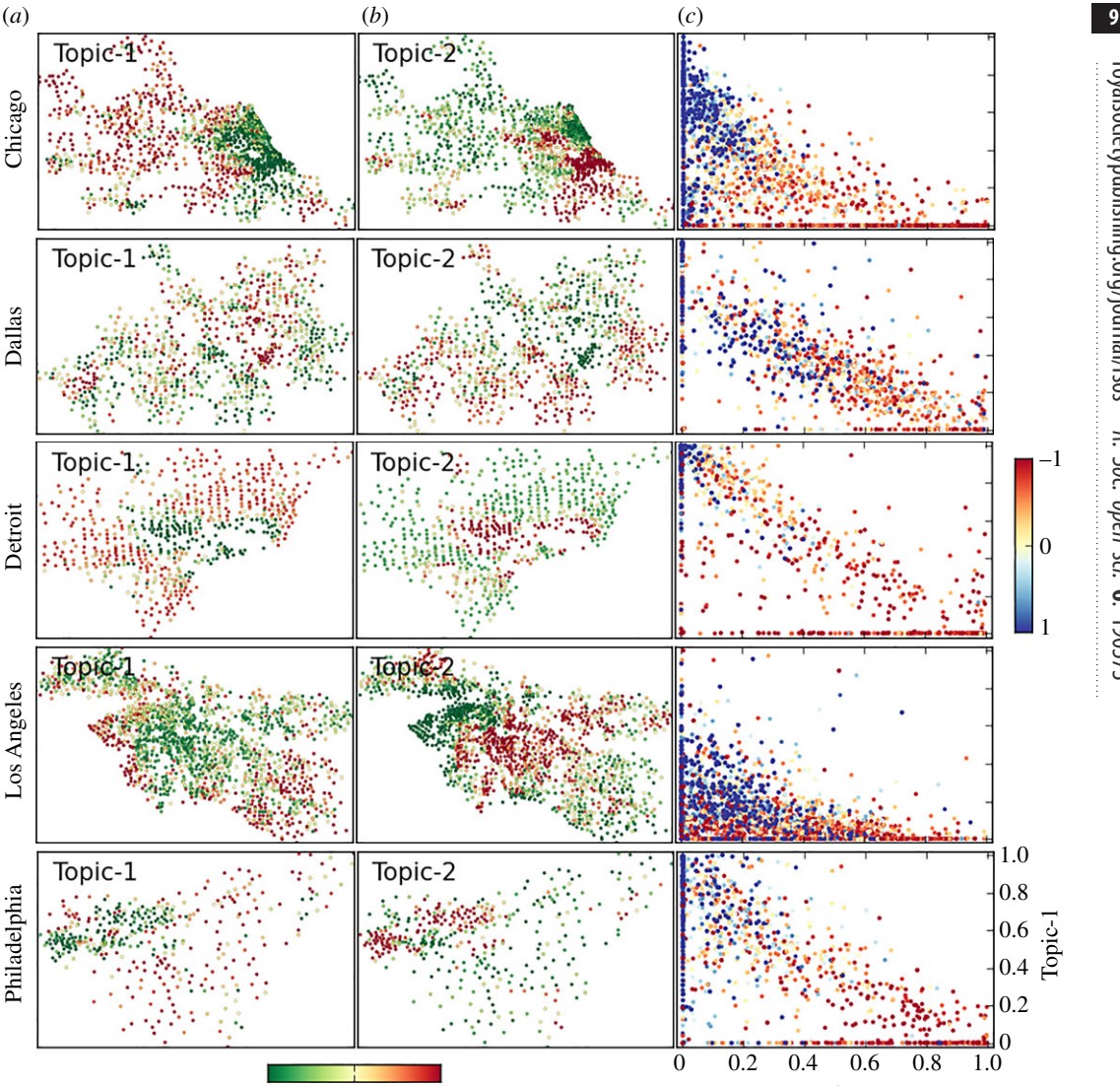

**Figure 6.** Polarization of discussion topics across neighbourhoods. Spatial distribution of topics on Twitter for Chicago, Dallas, Detroit, Los Angeles and Philadelphia (rows). (*a*) The dominant topic in wealthier areas (Topic-1). (*b*) The dominant topic in poorer areas (Topic-2). Dots indicate geolocated neighbourhoods, and colours indicate the normalized intensity of the cluster in each neighbourhood (colour scale shown in figure). (*c*) Scatter plots in which each neighbourhood is represented by a dot and the coordinates represent the extent of conversation of Topic-1 (vertical axis) and Topic-2 (horizontal axis) in that neighbourhood. The colour of each dot indicates the normalized median income (colour scale shown in figure).

plots in figure 7, correlations are annotated inset). Thus, exposure is the relevant variable for homogenization, and segregation simply constrains the space in which it happens. The element-wise correlation of the matrices after grouping elements by income ($r$) or random association ($r_r$) is shown in table 5 for all cities. $r_r$ is estimated from 100 realizations of the experiment of setting up neighbourhoods in groups by random association. We present the one-sided probability ($p$-value) that $r$ is the average of a normal distribution whose average and standard deviation are given by $r_r$.

A regression analysis shows that mutual exposure via mobility is a significant predictor of hashtag cosine distance between neighbourhoods across all seven cities, even after controlling for population, distance between neighbourhoods, and income. Specifically, we fit the data to the following model:

$$h_{ij} = \beta_0 + \beta_1 m_{ij} + \beta_2 d_{ij} + \beta_3 p_{ij} + \beta_4 w_{ij},$$

where $h_{ij}$ is the cosine similarity between hashtag usage vectors from neighbourhoods $i$ and $j$, $m_{ij}$ is the number of people living in neighbourhood $i$ that visit neighbourhood $j$, $d_{ij}$ is the distance between the

**Table 3.** Income topic correlation. We present the correlation of Topics 1 and 2 with income and the average correlation with standard error of randomly shuffled hashtag vectors among neighbourhoods. The probability of these correlations to be part of the random distribution is very low ($p < 0.001$).

| city | Topic-1 ($r > 0$) | Topic-2 ($r < 0$) | random ($r_r$) |
|---|---|---|---|
| Chicago | 0.59 | −0.59 | 0.002 ± 0.002 |
| Dallas | 0.31 | −0.35 | 0.001 ± 0.003 |
| Detroit | 0.57 | −0.52 | 0.003 ± 0.003 |
| Istanbul | 0.66 | −0.64 | 0.000 ± 0.004 |
| Los Angeles | 0.36 | −0.35 | −0.001 ± 0.002 |
| New York City | 0.32 | −0.42 | 0.001 ± 0.002 |
| Philadelphia | 0.48 | −0.54 | 0.000 ± 0.003 |

**Table 4.** Average distance of topics distribution to the city geographical centre. The distances are in kilometres. Topic-1 indicates topics that are more popular in wealthier areas. Topic-2 indicates topics that are more popular in poorer areas. See figures 5 and 6 for visualizations of the topics' spatial distribution.

| city | Topic-1 | Topic- 2 |
|---|---|---|
| Chicago | 28.33 | 23.94 |
| Dallas | 25.41 | 27.47 |
| Detroit | 20.59 | 12.47 |
| Istanbul | 12.77 | 15.47 |
| Los Angeles | 28.06 | 21.15 |
| New York City | 9.10 | 11.31 |
| Philadelphia | 14.97 | 11.32 |

centroids of neighbourhoods $i$ and $j$, $p_{ij}$ is the geometric-average Twitter population of neighbourhoods $i$ and $j$, and $w_{ij}$ is the geometric-average income of neighbourhoods $i$ and $j$. Twitter population represents the number of Twitter users whose home locations we are able to determine. Income information is taken from the Census (median Census tract income). Variables $h_{ij}$, $m_{ij}$, $p_{ij}$ and $w_{ij}$ have been normalized by subtracting the average and dividing by the standard deviation prior to running the regression. The regression coefficients and variance explained ($R^2$) are shown in table 6. The results show that for most cities the amount of inter-neighbourhood mobility is more important than the geographical distance between neighbourhoods.

Previous studies show that geographical distance constrains both social ties [30] and human mobility patterns [31], which has an effect on the segregation we observe due to the close location of neighbourhoods of similar income. However, the central/peripheral patterns of the topics of conversation in figures 5 and 6, as well as the results of the regression shown in table 6, indicate that distance is not the only factor and in most cases the inter-neighbourhood mobility seems to be more influential to determine the homogenization of behaviours.

# 3. Conclusion

In summary, urban segregation fractures the social space of mutual exposure, promoting polarization rather than global homogenization of behaviours. We provide direct observation over a large number of social interactions that the structure of interactions and spread of behaviours are consistent with each other, and that because cities are segregated places, information does not flow homogeneously across social classes in either the physical or virtual space. Although the Twitter datasets may have inherent biases, our results are in agreement with Census data, credit card data and weighted samples.

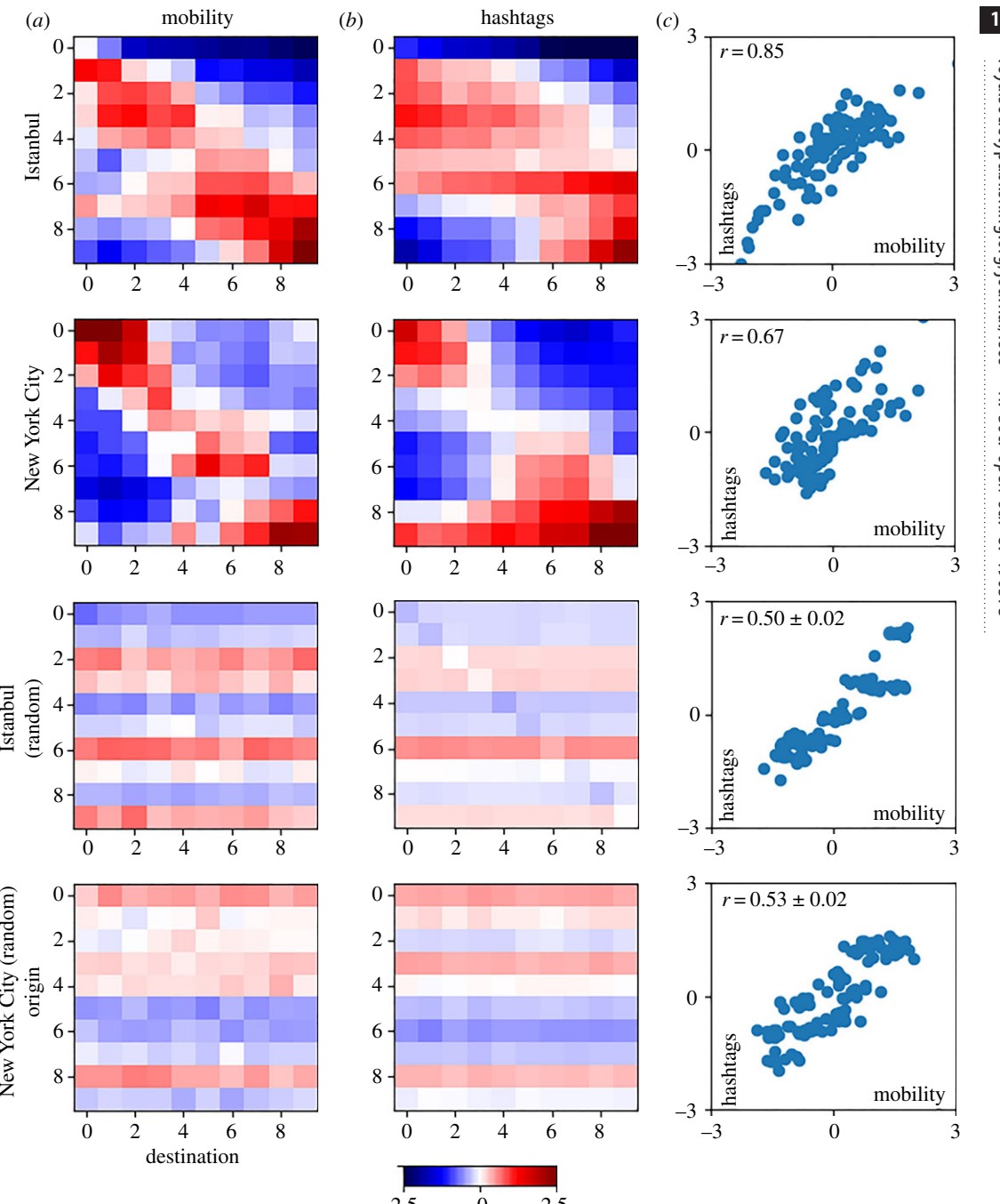

**Figure 7.** Inter-neighbourhood social exposure through mobility (*a*) and hashtag similarities (*b*), for Istanbul (first and third row) and New York City (second and fourth row). First and second rows are obtained after aggregating neighbourhoods by income, and third and fourth rows by random aggregation. Matrices have been normalized relative to the corresponding standard deviation from income aggregation (scale in figure). Scatter plots in (*c*) show the correlation (*r*) between mutual exposure through mobility and hashtag similarities for both types of aggregation. In the random case, we show the average correlation and standard error after 100 realizations. Results for more cities are presented in electronic supplementary material, S6.

Through segregation, people create different social spaces, where language, expressions and interests become idiosyncratic, within-group communication is effective and distinct identities emerge. That urban areas have multiple subcultures need not be considered a societal problem [24]. Difficulties can arise, however, when societal norms are not shared [32], especially, those relevant to services and resources, such as health or education. Such inconsistencies have created asymmetries and friction in multiple societies [33]. Accommodating multiple views is more readily possible if governance structures are aligned with group identities [34].

**Table 5.** Correlation between elements of matrices of human mobility and hashtag similarities among neighbourhoods. $r$ stands for correlation when neighbourhoods are grouped in 10 sets ordered by income. $r_r$ shows the average correlation and standard error of 100 realizations of the experiment of setting up neighbourhoods in 10 groups by random association. $p$-value shows the probability that $r$ is the average of a normal distribution with location and scale as $r_r$. The probability of these correlations to be part of the random distribution is very low ($p < 0.001$).

| city | $r$ | $r_r$ |
|---|---|---|
| Chicago | 0.74 | $0.57 \pm 0.02$ |
| Dallas | 0.70 | $0.48 \pm 0.02$ |
| Detroit | 0.82 | $0.45 \pm 0.02$ |
| Istanbul | 0.85 | $0.50 \pm 0.02$ |
| Los Angeles | 0.68 | $0.45 \pm 0.03$ |
| New York City | 0.67 | $0.53 \pm 0.02$ |
| Philadelphia | 0.89 | $0.45 \pm 0.02$ |

**Table 6.** Regression analysis of hashtag similarities between neighbourhoods as a function of: mobility between neighbourhoods ($\beta_1$), distance between neighbourhood centroids ($\beta_2$), geometric-average neighbourhood Twitter population ($\beta_3$) and geometric-average neighbourhood income ($\beta_4$). Parentheses show 95% confidence intervals. $R^2$ stands for the model explained variance. $\beta$ coefficients are significant ($p < 0.001$), with the exception of $\beta_4$ for Detroit which is not significant.

| city | $\beta_1$ (mobility) | $\beta_2$ (distance) | $\beta_3$ (population) | $\beta_4$ (income) | $R^2$ |
|---|---|---|---|---|---|
| Chicago | 0.161 | −0.215 | 0.422 | 0.147 | 0.37 |
| | (0.153, 0.168) | (−0.222, −0.207) | (0.414, 0.429) | (0.140, 0.154) | |
| Dallas | 0.363 | −0.073 | 0.248 | 0.039 | 0.24 |
| | (0.353, 0.373) | (−0.084, −0.063) | (0.238, 0.259) | (0.029, 0.048) | |
| Detroit | 0.240 | −0.039 | 0.213 | −0.008 | 0.13 |
| | (0.225, 0.255) | (−0.054, −0.024) | (0.198, 0.228) | (−0.021, 0.006) | |
| Istanbul | 0.116 | −0.042 | 0.601 | −0.063 | 0.40 |
| | (0.110, 0.122) | (−0.048, −0.036) | (0.595, 0.607) | (−0.069, −0.058) | |
| Los Angeles | 0.347 | −0.179 | 0.163 | 0.079 | 0.23 |
| | (0.340, 0.355) | (−0.186, −0.172) | (0.156, 0.171) | (0.072, 0.086) | |
| New York City | 0.094 | −0.336 | 0.430 | 0.113 | 0.36 |
| | (0.083, 0.106) | (−0.347, −0.325) | (0.419, 0.442) | (0.102, 0.124) | |
| Philadelphia | 0.215 | −0.140 | 0.273 | −0.044 | 0.18 |
| | (0.194, 0.235) | (−0.160, −0.120) | (0.252, 0.294) | (−0.064, −0.024) | |

Although it might be more difficult to change the nature of human behaviour, the spatial design of physical infrastructure can be used to influence the development of shared norms by fostering interdependencies. Desegregating basic human activities like work or trade could promote mixing and shared norms, and the right kinds of dependencies can be expected to reduce conflict. Evidence for this can, for instance, be found in historical interactions among nations [35]. However, achieving social integration when jobs are becoming increasingly segregated [36] remains a challenging, yet important, task in modern societies.

# 4. Methods

We collected over 87 million tweets from over 2.8 million users between August 2013 and August 2014 using the Stream Application Programming Interface (API), which provides over 90% of the publicly

**Table 7.** Coefficients of regression analysis to study biases of Twitter population. The dependent variable is the Twitter population. The independent variables are: Census population, median income, median age and male ratio. Samples represent neighbourhoods. All variables have been normalized by subtracting the average and dividing by the standard deviation. With the exception of the Twitter population, the data has been taken from the Census. Parentheses show 95% confidence intervals. $R^2$ stands for the model explained variance. Population and median age coefficients are significant ($p < 0.001$). Median income is significant for Chicago, Dallas and New York ($p < 0.001$) as well as for Los Angeles ($p < 0.1$). Median age is significant for all cities ($p < 0.001$) including Detroit ($p < 0.05$). Male ratio is only significant in Detroit and New York City ($p < 0.001$).

| city | population | median income | median age | male ratio | $R^2$ |
|---|---|---|---|---|---|
| Chicago | 0.55 | 0.11 | −0.19 | −0.02 | 0.35 |
| | (0.52, 0.59) | (0.07, 0.15) | (−0.23, −0.16) | (−0.05, 0.02) | |
| Dallas | 0.57 | 0.14 | −0.19 | −0.03 | 0.40 |
| | (0.52, 0.61) | (0.09, 0.19) | (−0.24, −0.14) | (−0.07, 0.02) | |
| Detroit | 0.63 | 0.01 | −0.06 | 0.12 | 0.41 |
| | (0.59, 0.68) | (−0.05, 0.6) | (−0.10, −0.01) | (0.08, 0.16) | |
| Los Angeles | 0.30 | 0.04 | −0.10 | −0.02 | 0.11 |
| | (0.27, 0.34) | (0.00, 0.09) | (−0.15, −0.06) | (−0.05, 0.02) | |
| New York City | 0.42 | 0.29 | −0.13 | 0.10 | 0.26 |
| | (0.39, 0.46) | (0.25, 0.33) | (−0.17, −0.09) | (0.06, 0.14) | |
| Philadelphia | 0.37 | −0.04 | −0.29 | −0.03 | 0.24 |
| | (0.32, 0.41) | (−0.09, 0.01) | (−0.34, −0.24) | (−0.07, 0.02) | |

available geolocated tweets [37]. Geolocated tweets provide a precise location of the individuals that post messages, and represent around 3% of the overall Twitter stream [38]. Its population trends younger and urban [39–41], which makes it a good probe of the dynamics of young workers in cities. The credit card purchase dataset was provided by a major financial institution in Istanbul. We analysed a total of 2.4 million records of individual credit card purchases in a three-month period in 2014, made by 85 000 individuals at 54 000 stores. All personal data have been anonymized following privacy laws and scientific standards (see electronic supplementary material, S1).

The shopping network is created by linking users' home neighbourhoods to the neighbourhoods where they shop, as determined from credit card transaction datasets. The mobility network is obtained by first analysing individual patterns of night-time Twitter activity to infer their home locations, and linking users' home neighbourhoods to the neighbourhoods they visit and tweet from. Similarly, the communications network is created by linking the home locations of users mentioning each other in their tweets. Neighbourhoods represent official sub-urban administrative units defined by national authorities: *mahalle* in Istanbul and *Census tract* in American cities. In the case of Chicago, Dallas, Detroit, Los Angeles and Philadelphia, we chose Census tracts associated to the Metropolitan Statistical Area (MSA). MSAs include both a substantial population nucleus, together with adjacent communities having a high degree of economic and social integration with that core. According to previous studies, Census tracts present a high variability in terms of income and racial distribution and therefore are suitable for studying segregation in urban areas [9]. For robustness, we chose neighbourhoods with at least five localizable Twitter users. The set of neighbourhoods analysed in each city remains unchanged across all experiments. We do not consider self-loops in the network analysis.

We check the robustness of our results against (i) Census data, (ii) balanced samples, and (iii) randomized versions of the datasets. Balanced samples are created by fixing under- and over-representation of neighbourhood's inhabitants in the Twitter sample, such that the neighbourhood sample size is proportional to the relative number of people by Census data with respect to the total city population. We randomize the datasets by shuffling the location of hashtag and mobility vectors across neighbourhoods, i.e. randomly assign the vector of one neighbourhood to another. The results are statistically significant and in agreement with balanced samples.

Biases in geolocated Twitter users have been previously analysed [39–41]. The under-age population is under-represented and biases have spatial patterns, such as a clear distinction between urban versus

rural areas. Despite biases, recent work has shown that the opinions collected on social media around relevant topics are not that different from traditional surveys [42].

In order to assess biases on our Twitter samples, we compared the number of Twitter users we identified living in each neighbourhood with demographics taken from Census data. We applied a regression analysis per each city. The dependent variable is the number of Twitter users per neighbourhood. The independent variables are: neighbourhood population, median income, median age and gender ratio. These data are taken from the Census. The coefficients are shown in table 7. In all cities, there is a clear bias towards younger population. In some cases, there is a smaller bias towards wealthier population. However, in the case of Detroit, Los Angeles and Philadelphia, income seems to be independent from the number of Twitter users. Gender seems to be irrelevant in most cases. Previous studies show that younger or wealthier people tend to explore the city more than older or poorer populations [43,44]. They also show that segregation is dynamic such that during night-time, weekends or winter there is less mixing than during office hours [45]. Additional correlations between Twitter and Census populations are shown in electronic supplementary material, S1.

Data accessibility. Data available from the Dryad Digital Repository: https://doi.org/10.5061/dryad.737m496 [46].

Authors' contributions. A.J.M carried out the data collection and data analysis, participated in the design of the study and drafted the manuscript; X.D. participated in the data analysis, design of the study and elaboration of the manuscript draft; Y.B.-Y. participated in the design of the study and helped writing the manuscript; A.P. participated in the design of the study, coordinated the analysis and helped drafting the manuscript. All authors gave final approval for publication.

Competing interests. We have no competing interests.

Funding. This manuscript has been funded by the Media Lab Member Consortium.

Acknowledgements. We thank Matthew Hardcastle for proofreading the manuscript.

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
