## [Reviewer comments · Royal Society Open Science]

Review History

RSOS-190573.R0 (Original submission)

Review form: Reviewer 1

Is the manuscript scientifically sound in its present form?

Yes

Are the interpretations and conclusions justified by the results?

Yes

Is the language acceptable?

Yes

Is it clear how to access all supporting data?

Yes

Do you have any ethical concerns with this paper?

No

Have you any concerns about statistical analyses in this paper?

No

Recommendation?

Accept with minor revision (please list in comments)

Comments to the Author(s)

This is a very interesting study, with novel findings regarding the levels of, and links between, income segregation, mutual exposure through mobility, and cultural homogenization as measured by twitter hashtag usage in several large U.S. cities and Istanbul. The findings are well supported by the data and analysis. I recommend this paper for publication, pending minor revisions that elaborate the limits of the data, deepen the discussion of the differences found between cities, ground the discussion of the causes and consequences of income segregation in a more solid review of the literature, and make more explicit the assumptions and choices made regarding geographic scale.

Segregation

The discussion of income segregation is narrowly blinkered by a methodological individualism that constructs segregation largely as a product of individual choice/preference. The literature on segregation by race and income (which are linked) shows that there are historical, institutional, structural, and infrastructural determinants of segregation that constrain and incentivize the choices available to individuals regarding their places of residence. I recommend deepening this discussion with reference to the work of scholars like William Julius Wilson (2008) and Swanstrom et. al. (2002). These authors discuss the social and individual damage that segregation does. This damage is not done mainly indirectly through the pathways of subculture formation that the authors note in their conclusion, but chiefly directly in ways which can be measured individually in terms of income, education, social mobility, health, lifespan, as well as socially in terms of social and environmental justice (Pulido 2000), community and democracy (Young 2002, Ch. 6).

Scale

There should be some justification for, and discussion of the limits of, the scale selected for the units of comparison and the neighborhood units. Why Census Tracts and not Block Groups? Why cities by municipal boundaries and not MSAs (which encompass the “commutershed” and would have captured many people who work in the city and live outside, and vice versa)? Neil Smith’s work on scale from a conceptual vantage and Fowler’s on scale and segregation from a methodological vantage may be useful here.

Comparison of Cities

I found the network graphs comparing the cities in Figures 1 and 5 fascinating. They prompt all sorts of speculation regarding the different patterns found among the cities studied. Insights from authors like Wilson, Swanstrom, and Pulido would prove useful here in explaining historically and structurally the differences among the patterns of segregation in the cities, which might provide grounds for a deeper, more specific and more useful set of conclusions regarding what can be done about segregation in the physical and virtual realms. Discussion of other differences between cities such as overall metrics of income segregation, presence or absence of public transit, clusters of employment in media and entertainment would be stimulating and help explain why New York and LA seem to be outliers in Table 2. And why LA appears to be an outlier in terms of polarization of discussion topics. And Detroit and NY in table 9 (is it NY’s subway system? Detroit’s poverty? Would a map that included the wealthy suburbs in the Detroit MSA tell a different story?).

It would also be very interesting to see a measure boiling down an overall level of clustering of mobility and communication for each city. We can tell by looking at the graphs that LA and Dallas have more interactions overall across income lines than other cities do-- but how much

more? And how much do these aggregate interaction/isolation levels differ between mobility and communication for each city?

Data and Presentation

-Some more details on the skews of the set of twitter users is called for (Page 15 line 32/33). What differences are known about patterns of mobility along lines of age, income, gender that might help us compensate for the limits of the twitter sample? Mei-Po Kwan's work on mobility and identity through the course of the day and workweek can be useful here. Are the poor people entering wealthy neighborhoods domestic servants? Further study contrasting the weekend to the weekday, and work hours to evening hours would be an interesting followup study.

-It would be helpful to know the breakpoints for the income deciles in each city-- ideally in the graphs themselves, i.e "\$35k" instead of "3" and so on. But if space and design prove too challenging, then at least as a table.

-The use of the rainbow color ramp in Figure 3 makes the maps pretty but sacrifices legibility of the overall pattern of the data. A two-color ramp would be more legible given that the data are bi-polar.

S. Fowler, C. (2016). Segregation as a multiscalar phenomenon and its implications for neighborhood-scale research: The case of South Seattle 1990–2010. *Urban geography*, 37(1), 1-25.

Pulido, L. (2000). Rethinking environmental racism: White privilege and urban development in Southern California. *Annals of the Association of American Geographers*, 90(1), 12-40.

Smith, N. (1992). Contours of a spatialized politics: Homeless vehicles and the production of geographical scale. *Social text*, (33), 55-81.

Swanstrom, T., Dreier, P., & Mollenkopf, J. (2002). Economic inequality and public policy: The power of place. *City & Community*, 1(4), 349-372.

Wilson, W. J. (2008). The political and economic forces shaping concentrated poverty. *Political Science Quarterly*, 123(4), 555-571.

Young, I. M. (2002). *Inclusion and democracy*. Oxford University press

Review form: Reviewer 2

Is the manuscript scientifically sound in its present form?

No

Are the interpretations and conclusions justified by the results?

No

Is the language acceptable?

Yes

Is it clear how to access all supporting data?

Yes

Do you have any ethical concerns with this paper?

No

Have you any concerns about statistical analyses in this paper?

No

Recommendation?

Major revision is needed (please make suggestions in comments)

Comments to the Author(s)

The article studies the segregation by income of individuals in the online and offline world. Relying mostly on the analysis of Twitter data (and partially on credit card transactions data), the authors show that individuals coming from neighborhoods with similar income are more likely to interact on Twitter and to discuss similar topics on Twitter. Moreover, they show that individuals coming from a neighborhood with a given income are more likely to be found in a neighborhood where individuals have similar income.

The topic under investigation is timely and interesting, and it presents interesting results on the connection between how we experience the online and offline world. The article is clearly written and suitable for the broad readership of Royal Society Open Science. Results are well presented, analysis and statistical tests are performed rigorously.

My main concern about the paper is that the authors do not consider one important confounding factor that can explain to a large extent the observed correlations: namely the distance between neighborhoods. It is well known that geographical distance constrain social ties (see for example <https://www.aaai.org/ocs/index.php/ICWSM/ICWSM11/paper/viewFile/2751/3307>) and mobility (<http://barabasi.com/f/250.pdf>), and due to spatial segregation, neighborhood with similar income are more likely to be located close to one another.

Therefore, claims such as “online interactions are segregated by income just as physical interactions are” are not fully supported by the current analysis. A possible way to go would be to run more realistic randomization, where the probability for individuals from two neighborhoods to interact is not uniform (this is a very unrealistic assumption).

I would like to see how the effects of social exposure/mobility are mediated by distance and to which extent income differences explain something that geographical distance does not.

I judge the paper will be suitable for publication once additional analyses show that the results can not be simply explained by geographical distance between neighbourhoods.

More detailed comments:

→ Figure 1: Specify what “communication on Twitter” means (retweet?)

→ Figure 1: The labels origin/destination are a bit confusing.

→ Do you remove individuals’ own neighbourhood from the analysis? (Individuals are more likely to be found in their own neighborhood and interact with individuals from their own neighborhood).

Decision letter (RSOS-190573.R0)

03-Jul-2019

Dear Dr Morales,

The editors assigned to your paper ("Segregation and Polarization in Urban Areas") have now received comments from reviewers. We would like you to revise your paper in accordance with the referee and Associate Editor suggestions which can be found below (not including

confidential reports to the Editor). Please note this decision does not guarantee eventual acceptance.

Please submit a copy of your revised paper before 26-Jul-2019. Please note that the revision deadline will expire at 00.00am on this date. If we do not hear from you within this time then it will be assumed that the paper has been withdrawn. In exceptional circumstances, extensions may be possible if agreed with the Editorial Office in advance. We do not allow multiple rounds of revision so we urge you to make every effort to fully address all of the comments at this stage. If deemed necessary by the Editors, your manuscript will be sent back to one or more of the original reviewers for assessment. If the original reviewers are not available, we may invite new reviewers.

- Data accessibility

<http://datadryad.org/submit?journalID=RSOS&manu=RSOS-190573>

- Competing interests

- Authors' contributions

- Acknowledgements

- Funding statement

on behalf of Dr Hamed Haddadi (Associate Editor) and Miles Padgett (Subject Editor)
openscience@royalsociety.org

Associate Editor's comments (Dr Hamed Haddadi):

Thank for this interesting submission. Please see the detailed reviewers' comments and update the paper accordingly for a resubmission.

Comments to Author:

Reviewers' Comments to Author:

Reviewer: 1

Comments to the Author(s)

This is a very interesting study, with novel findings regarding the levels of, and links between, income segregation, mutual exposure through mobility, and cultural homogenization as measured by twitter hashtag usage in several large U.S. cities and Istanbul. The findings are well

supported by the data and analysis. I recommend this paper for publication, pending minor revisions that elaborate the limits of the data, deepen the discussion of the differences found between cities, ground the discussion of the causes and consequences of income segregation in a more solid review of the literature, and make more explicit the assumptions and choices made regarding geographic scale.

Segregation

The discussion of income segregation is narrowly blinkered by a methodological individualism that constructs segregation largely as a product of individual choice/preference. The literature on segregation by race and income (which are linked) shows that there are historical, institutional, structural, and infrastructural determinants of segregation that constrain and incentivize the choices available to individuals regarding their places of residence. I recommend deepening this discussion with reference to the work of scholars like William Julius Wilson (2008) and Swanstrom et. al. (2002). These authors discuss the social and individual damage that segregation does. This damage is not done mainly indirectly through the pathways of subculture formation that the authors note in their conclusion, but chiefly directly in ways which can be measured individually in terms of income, education, social mobility, health, lifespan, as well as socially in terms of social and environmental justice (Pulido 2000), community and democracy (Young 2002, Ch. 6).

Scale

There should be some justification for, and discussion of the limits of, the scale selected for the units of comparison and the neighborhood units. Why Census Tracts and not Block Groups? Why cities by municipal boundaries and not MSAs (which encompass the “commutershed” and would have captured many people who work in the city and live outside, and vice versa)? Neil Smith’s work on scale from a conceptual vantage and Fowler’s on scale and segregation from a methodological vantage may be useful here.

Comparison of Cities

I found the network graphs comparing the cities in Figures 1 and 5 fascinating. They prompt all sorts of speculation regarding the different patterns found among the cities studied. Insights from authors like Wilson, Swanstrom, and Pulido would prove useful here in explaining historically and structurally the differences among the patterns of segregation in the cities, which might provide grounds for a deeper, more specific and more useful set of conclusions regarding what can be done about segregation in the physical and virtual realms. Discussion of other differences between cities such as overall metrics of income segregation, presence or absence of public transit, clusters of employment in media and entertainment would be stimulating and help explain why New York and LA seem to be outliers in Table 2. And why LA appears to be an outlier in terms of polarization of discussion topics. And Detroit and NY in table 9 (is it NY’s subway system? Detroit’s poverty? Would a map that included the wealthy suburbs in the Detroit MSA tell a different story?).

It would also be very interesting to see a measure boiling down an overall level of clustering of mobility and communication for each city. We can tell by looking at the graphs that LA and Dallas have more interactions overall across income lines than other cities do-- but how much more? And how much do these aggregate interaction/isolation levels differ between mobility and communication for each city?

Data and Presentation

-Some more details on the skews of the set of twitter users is called for (Page 15 line 32/33). What differences are known about patterns of mobility along lines of age, income, gender that might help us compensate for the limits of the twitter sample? Mei-Po Kwan’s work on mobility and

identity through the course of the day and workweek can be useful here. Are the poor people entering wealthy neighborhoods domestic servants? Further study contrasting the weekend to the weekday, and work hours to evening hours would be an interesting followup study.

-It would be helpful to know the breakpoints for the income deciles in each city-- ideally in the graphs themselves, i.e. "\$35k" instead of "3" and so on. But if space and design prove too challenging, then at least as a table.

-The use of the rainbow color ramp in Figure 3 makes the maps pretty but sacrifices legibility of the overall pattern of the data. A two-color ramp would be more legible given that the data are bi-polar.

S. Fowler, C. (2016). Segregation as a multiscale phenomenon and its implications for neighborhood-scale research: The case of South Seattle 1990–2010. *Urban geography*, 37(1), 1-25.

Pulido, L. (2000). Rethinking environmental racism: White privilege and urban development in Southern California. *Annals of the Association of American Geographers*, 90(1), 12-40.

Smith, N. (1992). Contours of a spatialized politics: Homeless vehicles and the production of geographical scale. *Social text*, (33), 55-81.

Swanstrom, T., Dreier, P., & Mollenkopf, J. (2002). Economic inequality and public policy: The power of place. *City & Community*, 1(4), 349-372.

Wilson, W. J. (2008). The political and economic forces shaping concentrated poverty. *Political Science Quarterly*, 123(4), 555-571.

Young, I. M. (2002). *Inclusion and democracy*. Oxford University press

Reviewer: 2

Comments to the Author(s)

The article studies the segregation by income of individuals in the online and offline world. Relying mostly on the analysis of Twitter data (and partially on credit card transactions data), the authors show that individuals coming from neighborhoods with similar income are more likely to interact on Twitter and to discuss similar topics on Twitter. Moreover, they show that individuals coming from a neighborhood with a given income are more likely to be found in a neighborhood where individuals have similar income.

The topic under investigation is timely and interesting, and it presents interesting results on the connection between how we experience the online and offline world.

The article is clearly written and suitable for the broad readership of Royal Society Open Science. Results are well presented, analysis and statistical tests are performed rigorously.

My main concern about the paper is that the authors do not consider one important confounding factor that can explain to a large extent the observed correlations: namely the distance between neighborhoods. It is well known that geographical distance constrain social ties (see for example <https://www.aaai.org/ocs/index.php/ICWSM/ICWSM11/paper/viewFile/2751/3307>) and mobility (<http://barabasi.com/f/250.pdf>), and due to spatial segregation, neighborhood with similar income are more likely to be located close to one another.

Therefore, claims such as "online interactions are segregated by income just as physical interactions are" are not fully supported by the current analysis. A possible way to go would be to run more realistic randomization, where the probability for individuals from two neighborhoods to interact is not uniform (this is a very unrealistic assumption).

I would like to see how the effects of social exposure/mobility are mediated by distance and to which extent income differences explain something that geographical distance does not.

I judge the paper will be suitable for publication once additional analyses show that the results can not be simply explained by geographical distance between neighbourhoods.

More detailed comments:

→ Figure 1: Specify what “communication on Twitter” means (retweet?)

→ Figure 1: The labels origin/destination are a bit confusing.

→ Do you remove individuals’ own neighbourhood from the analysis? (Individuals are more likely to be found in their own neighborhood and interact with individuals from their own neighborhood).

Author's Response to Decision Letter for (RSOS-190573.R0)

See Appendix A.

RSOS-190573.R1 (Revision)

Review form: Reviewer 1

Is the manuscript scientifically sound in its present form?

Yes

Are the interpretations and conclusions justified by the results?

Yes

Is the language acceptable?

Yes

Do you have any ethical concerns with this paper?

No

Have you any concerns about statistical analyses in this paper?

No

Recommendation?

Accept as is

Comments to the Author(s)

Thorough and thoughtful edits that respond to all suggestions. The only final suggestion I have is that you explain somewhere why the analysis for New York City is based on the city limits whereas the other U.S. cities are analyzed based on MSAs.

Review form: Reviewer 2

Is the manuscript scientifically sound in its present form?

Yes

Are the interpretations and conclusions justified by the results?

Yes

Is the language acceptable?

Yes

Do you have any ethical concerns with this paper?

No

Have you any concerns about statistical analyses in this paper?

No

Recommendation?

Accept as is

Comments to the Author(s)

I would like to congratulate the authors for improving the manuscript.
My comments have been addressed.

Decision letter (RSOS-190573.R1)

23-Sep-2019

Dear Dr Morales,

I am pleased to inform you that your manuscript entitled "Segregation and Polarization in Urban Areas" is now accepted for publication in Royal Society Open Science.

on behalf of Prof Miles Padgett (Subject Editor)
openscience@royalsociety.org

Reviewer comments to Author:

Reviewer: 1

Comments to the Author(s)

Thorough and thoughtful edits that respond to all suggestions. The only final suggestion I have is that you explain somewhere why the analysis for New York City is based on the city limits whereas the other U.S. cities are analyzed based on MSAs.

Reviewer: 2

Comments to the Author(s)

I would like to congratulate the authors for improving the manuscript.
My comments have been addressed.

Appendix A

Comments to Author:

Reviewers' Comments to Author:

Reviewer: 1

Comments to the Author(s)

This is a very interesting study, with novel findings regarding the levels of, and links between, income segregation, mutual exposure through mobility, and cultural homogenization as measured by twitter hashtag usage in several large U.S. cities and Istanbul. The findings are well supported by the data and analysis. I recommend this paper for publication, pending minor revisions that

- elaborate the limits of the data,

We extended the discussion on the limits of Twitter data in the Methods section and included a new analysis to measure biases in our samples across American neighborhoods in new Table 7. The new analysis consists on a linear regression model applied to each city, where the dependent variable is the number of Twitter users per census tract and the independent variables include Census population, median age, median income and sex ratio. We show that Twitter population trends younger in most cases, wealthier in very few cases, and not influenced by gender in any case.

- deepen the discussion of the differences found between cities,

We moved previous figures 5 and 25, as well as previous tables 4, 5 and 8 to the main text. We include new analyses: (1) one in which we calculate the extent of segregation in each city in new Figure 3 and new Table 1, (2) another comparing the spatial distribution of the topics of conversation across cities in new Table 4, (3) a regression showing the factors that influence content homogenization across neighborhoods in new Table 6 and (4) a regression analysis to compare demographic biases on Twitter samples in new Table 7.

In new Table 4 we show that the historic migration of richer people towards suburbs in most american urban areas (MSA). New York City shows a similar behavior as Istanbul, since wealthier people live near the city downtown rather than far from it. is reflected in the differentiation of topics of conversation on Twitter.

- ground the discussion of the causes and consequences of income segregation in a more solid review of the literature,

We included a new paragraph based on the papers recommended by the reviewer explaining the causes and negative consequences of segregation in American cities (new references 8-13)

- make more explicit the assumptions and choices made regarding geographic scale.

We added new sentences explaining that we have used MSA to define the geographic scale and based our choice on the variability found at the census tracts level according to the paper recommended by the reviewer.

Segregation

The discussion of income segregation is narrowly blinkered by a methodological individualism that constructs segregation largely as a product of individual choice/preference. The literature on segregation by race and income (which are linked) shows that there are historical, institutional, structural, and infrastructural determinants of segregation that constrain and incentivize the choices available to individuals regarding their places of residence. I recommend deepening this discussion with reference to the work of scholars like William Julius Wilson (2008) and Swanstrom et. al. (2002). These authors discuss the social and individual damage that segregation does. This damage is not done mainly indirectly through the pathways of subculture formation that the authors note in their conclusion, but chiefly directly in ways which can be measured individually in terms of income, education, social mobility, health, lifespan, as well as socially in terms of social and environmental justice (Pulido 2000), community and democracy (Young 2002, Ch. 6).

As the reviewer suggested, the literature shows that the distribution of neighborhood race and income influenced the evolution of American cities. We have included a discussion with citations to the recommended papers (3rd paragraph), where we outline the negative effects of segregation on social groups according to the aspects raised by the referee.

Scale

There should be some justification for, and discussion of the limits of, the scale selected for the units of comparison and the neighborhood units. Why Census Tracts and not Block Groups? Why cities by municipal boundaries and not MSAs (which encompass the “commutershed” and would have captured many people who work in the city and live outside, and vice versa)? Neil Smith’s work on scale from a conceptual vantage and Fowler’s on scale and segregation from a methodological vantage may be useful here.

In the case of Chicago, Dallas, Detroit, Los Angeles and Philadelphia we have chosen MSAs as the scale of observation. They contain both a nucleus of substantial population density together with the adjacent communities that hold economic and social ties with that nucleus. We analyze segregation using Census Tracts because they show high variability according to Fowler’s study. As we aggregate by income to create the interaction matrices, the individual heterogeneities are averaged and the collective behavior is manifested. According to the science of complex systems, emergent phenomena are driven by the collective behaviors. We have included the description and justification of the scale used in section Methods.

Comparison of Cities

I found the network graphs comparing the cities in Figures 1 and 5 fascinating. They prompt all sorts of speculation regarding the different patterns found among the cities studied. Insights from authors like Wilson, Swanstrom, and Pulido would prove useful here in explaining historically and structurally the differences among the patterns of segregation in the cities, which might provide grounds for a deeper, more specific and more useful set of conclusions regarding what can be done about segregation in the physical and virtual realms.

We thank the reviewer for the suggestion. We have moved Figure 5 to the main text (new Figure 2) and included a new discussion to the papers recommended by the reviewer.

Discussion of other differences between cities such as overall metrics of income segregation, presence or absence of public transit, clusters of employment in media and entertainment would be stimulating and help explain why New York and LA seem to outliers in Table 2. And how much do these aggregate interaction/isolation levels differ between mobility and communication for each city? And why LA appears to be an outlier in terms of polarization of discussion topics. And Detroit and NY in table 9 (is it NY's subway system? Detroit's poverty? Would a map that included the wealthy suburbs in the Detroit MSA tell a different story?). It would also be very interesting to see a measure boiling down an overall level of clustering of mobility and communication for each city. We can tell by looking at the graphs that LA and Dallas have more interactions overall across income lines than other cities do-- but how much more?

In order to expand the comparison among cities we have moved multiple figures and tables from the supplement to the main text and show the behavior of all cities in every aspect. We have also included new analyses (explained below) to measure the level of segregation in each city as well as differences in the spatial distribution of the topics of conversation. As the reviewer suggested, the history of policies and market behaviors that shaped the evolution of urban developments in America is manifested in many of the observed patterns of social media communication.

We included a new analysis measuring the distance of the matrices in old Figures 1 and 5 (new Figures 1 and 2) with respect to the uniform distribution. The analysis includes one new figure showing the Q-Q plots (new Figure 3), a table with the test results (new Table 1), and a new paragraph of discussion in the main text. We show that the level of clustering using mobility data is lower than the level of clustering using online communication. Also, we show that cities like Istanbul and LA show less segregation in terms of mobility than other cities.

We also included a new analysis of the topics of conversation. We moved the previous Figure 25 to the main text (new Figure 6) where we show the topics of wealthier and poorer communities in the additional 5 American cities. The new analysis (new Table 4) includes measuring the average distance of topics to the geographic city center. We found that most of American cities show a different behavior than Istanbul or NYC given the particular settlement patterns of the former cities that evolved as described by the reviewer's recommended papers which have also been included in the discussion.

Data and Presentation

-Some more details on the skews of the set of twitter users is called for (Page 15 line 32/33). What differences are known about patterns of mobility along lines of age, income, gender that might help us compensate for the limits of the twitter sample? Mei-Po Kwan's work on mobility and identity through the course of the day and workweek can be useful here. Are the poor people entering wealthy neighborhoods domestic servants? Further study contrasting the weekend to the weekday, and work hours to evening hours would be an interesting followup study.

We added a new regression analysis (new Table 7) to estimate biases in our samples based on age, income and gender as the reviewer suggested. We applied one regression per city. The dependent variable is the number of Twitter users per neighborhood. The independent variables are: neighborhood population, median income, median age and gender ratio. The data is taken from the Census. The coefficients are shown in new Table 7. We found a tendency for younger population in most cities. Only some cities show a tendency of wealthier population. Gender seems not to be relevant for determining Twitter population.

We also included the following references in the discussion of human mobility:

Mei-Po Kwan and Tim Schwanen. Geographies of mobility. *Annals of the American Association of Geographers*, 106(2):243–256, 2016.

S. Isaacman, R. Becker, R. Caceres, S. Kobourov, M. Martonosi, J. Rowland, and A. Varshavsky. Ranges of human mobility in los angeles and new york. In 2011 IEEE International Conference on Pervasive Computing and Communications Workshops , pages 88–93, March 2011.

Feixiong Luo, Guofeng Cao, Kevin Mulligan, and Xiang Li. . Explore spatiotemporal and demographic characteristics of human mobility via Twitter: A case study of Chicago. *Applied Geography*, 70:11 – 25, 2016

-It would be helpful to know the breakpoints for the income deciles in each city-- ideally in the graphs themselves, i.e "\$35k" instead of "3" and so on. But if space and design prove too challenging, then at least as a table.

We added a new Table with the income range per quantile for each city in the Supplement (Section S1, Table S3) and a reference to this table in the main text.

-The use of the rainbow color ramp in Figure 3 makes the maps pretty but sacrifices legibility of the overall pattern of the data. A two-color ramp would be more legible given that the data are bi-polar.

We changed the color scheme to a diverging color map both in new Figures 5 and 6 (previously 3 and 25).

Finally, as suggested by the reviewer, we have included the following references:

- S. Fowler, C. (2016). Segregation as a multiscale phenomenon and its implications for neighborhood-scale research: The case of South Seattle 1990–2010. *Urban geography*, 37(1), 1-25.
- Pulido, L. (2000). Rethinking environmental racism: White privilege and urban development in Southern California. *Annals of the Association of American Geographers*, 90(1), 12-40.
- Smith, N. (1992). Contours of a spatialized politics: Homeless vehicles and the production of geographical scale. *Social text*, (33), 55-81.
- Swanstrom, T., Dreier, P., & Mollenkopf, J. (2002). Economic inequality and public policy: The power of place. *City & Community*, 1(4), 349-372.
- Wilson, W. J. (2008). The political and economic forces shaping concentrated poverty. *Political Science Quarterly*, 123(4), 555-571.
- Young, I. M. (2002). *Inclusion and democracy*. Oxford University press

Reviewer: 2

Comments to the Author(s)

The article studies the segregation by income of individuals in the online and offline world. Relying mostly on the analysis of Twitter data (and partially on credit card transactions data), the authors show that individuals coming from neighborhoods with similar income are more likely to interact on Twitter and to discuss similar topics on Twitter. Moreover, they show that individuals coming from a neighborhood with a given income are more likely to be found in a neighborhood where individuals have similar income.

The topic under investigation is timely and interesting, and it presents interesting results on the connection between how we experience the online and offline world.

The article is clearly written and suitable for the broad readership of Royal Society Open Science. Results are well presented, analysis and statistical tests are performed rigorously.

My main concern about the paper is that the authors do not consider one important confounding factor that can explain to a large extent the observed correlations: namely the distance between neighborhoods. It is well known that geographical distance constrain social ties (see for example <https://www.aaai.org/ocs/index.php/ICWSM/ICWSM11/paper/viewFile/2751/3307>) and mobility (<http://barabasi.com/f/250.pdf>), and due to spatial segregation, neighborhood with similar income are more likely to be located close to one another.

We moved a regression analysis from the supplement to the main text (new Table 6) where we show that mobility is a predictor of hashtag similarities among neighborhoods after controlling for distance between neighborhoods among other factors. We have also expanded the discussion and included the references suggested by the reviewer (references [30,31]).

Therefore, claims such as “online interactions are segregated by income just as physical interactions are” are not fully supported by the current analysis. A possible way to go would be to run more realistic randomization, where the probability for individuals from two neighborhoods to interact is not uniform (this is a very unrealistic assumption).

A common way to measure segregation in the literature is based on the entropy of the racial or income distribution in neighborhoods (see reference [9]). Entropy is a measure of diversity. The maximum value is given by the uniform distribution, where the probability of all kinds is equal and the population is highly diverse. The minimum value is given by the Dirac Delta, where only one type accounts for 100% of the population. Based on these two extreme cases, we analyze the segregation of interactions according to income quantiles. We show both visually and quantitatively that conditional on the source of the interaction, the probability of the target is biased towards the source's income. We measure the distance from the uniform distribution using QQ plots (see Figure 3 and Table 1) and KS tests (Table S4).

I would like to see how the effects of social exposure/mobility are mediated by distance and to which extent income differences explain something that geographical distance does not. I judge the paper will be suitable for publication once additional analyses show that the results can not be simply explained by geographical distance between neighbourhoods.

We have included a new regression analysis (Table 6) that shows that distance is not the main factor to explain the effects of social exposure on the emergent behavior. We also show by means of pair matching that the emergent behaviors of content similarities have a consistent structure with the segregation of interactions both online and offline (Figure 7, Figure S23 and Table 5). Moreover, the central/peripheral patterns shown in Figures 5 and 6 demonstrate that is not just distance what matters for influencing the behavior of the population.

More detailed comments:

→ Figure 1: Specify what “communication on Twitter” means (retweet?)

We included in the main text and Methods section that communication is measured via the mentions mechanism.

→ Figure 1: The labels origin/destination are a bit confusing.

We changed the terms to source/target of the interaction

→ Do you remove individuals' own neighbourhood from the analysis? (Individuals are more likely to be found in their own neighborhood and interact with individuals from their own neighborhood).

Yes we do remove self-loops in the network analysis. We have included an explicit sentence in the Methods section to clarify this action.